# Social Media Activities, Emotion Regulation Strategies, and Their Interactions on People’s Mental Health in COVID-19 Pandemic

**DOI:** 10.3390/ijerph17238931

**Published:** 2020-12-01

**Authors:** Yang Yang, Keqiao Liu, Siqi Li, Man Shu

**Affiliations:** 1Institute of Advanced Studies in Humanities and Social Sciences, Beijing Normal University at Zhuhai, Zhuhai 519087, China; yyang37@buffalo.edu; 2Research Institute of Science Education, Beijing Normal University, Beijing 100875, China; 3School of Public Finance and Public Administration, Jiangxi University of Finance and Economics, Nanchang 330013, China; Keqiaoliu@outlook.com; 4College of Education for the Future, Beijing Normal University at Zhuhai, Zhuhai 519087, China; 5Institute of Psychological Quality and Education, East China Jiaotong University, Nanchang 330013, China

**Keywords:** social media activities, emotion-regulation strategies, mental health, COVID-19 pandemic

## Abstract

The COVID-19 pandemic has dramatically changed the general population’s life worldwide. People may spend more time on social media because of policies like “work at home”. Using a cross-sectional dataset collected through an online survey in February 2020, in China, we examined (1) the relationships between social media activities and people’s mental health status and (2) the moderation effect of emotional-regulation strategies. The sample included people aged ≥18 years from 32 provinces and regions in China (*N* = 3159). The inferential analyses included a set of multiple linear regressions with interactions. Our results showed that sharing timely, accurate, and positive COVID-19 information, reducing excessive discussions on COVID-19, and promoting caring online interactions rather than being judgmental, might positively associate with the general public’s psychological well-being. Additionally, the relationships between social media activities and psychological well-being varied at different emotion-regulation strategy levels. Adopting the cognitive reappraisal strategy might allay the adverse relationships between certain social media activities and mental health indicators. Our findings expanded the theory of how social media activities can be associated with a human being’s mental health and how it can interact with emotion-regulation strategies during the COVID-19 pandemic.

## 1. Introduction

On 11 March, the WHO officially declared the COVID-19 as a pandemic. As of 8 September 2020, it has spread to 216 countries, with over 27 million confirmed cases around the world [1]. To combat the COVID-19, the Chinese government implemented strict policies for a massive lockdown, as well as tracking and isolating suspect cases for the first time, which significantly lowered the confirmed cases within two months. Governments of other countries also have imposed public control policies, such as working-from-home solutions, as well as travel and gathering restrictions. Currently, the “desperate flea” [2] phenomena have become universal around the world, where many people stay at home and are socially isolated. A number of studies have pointed out that pandemic [3,4] and social isolation [5,6] could negatively influence the mental health of the general population. For example, people’s worry of the swine flu pandemic was positively associated with the volume of media reporting (of the pandemic), after the outbreak of the flu in the UK [4]. Therefore, a timely understanding of the psychological manifestations of the general public is imperatively needed for the society during the COVID-19 pandemic. Specifically, this study aims to understand the associations of social media activities, emotion-regulation strategies, and their interactions with the general populations’ mental health under the COVID-19 pandemic.

### 1.1. Mental Health of Chinese People during the COVID-19 Pandemic

The general population in China has experienced the unprecedented strict “stay at home” policy from the beginning of the COVID-19 outbreak. Research on Chinese people’s psychological manifestations (e.g., anxiety, depression, happiness, sleep quality, and stress) during the pandemic could provide insights for research and practice related to the pandemic worldwide. A large amount of those studies reported higher anxiety and depression levels [7,8,9,10,11,12], increased stress [13], and lower life-satisfaction levels among the Chinese general population [9,13,14] since the outbreak of COVID-19. However, Zhang and Ma [15] reported a mild stressful impact of COVID-19 on their sample, shortly after the outbreak, which, as suggested by the researcher, might relate to reasons including sample characteristics. Similarly, a longitudinal study indicated that, despite of the sharp increases in COVID-19-confirmed cases between the end of January (10 days after the outbreak) and the beginning of March, the levels of stress, anxiety, and depression were relatively stable among the general population [11].

Researchers further explored a number of factors that were associated with psychological manifestations. They found the following: (1) Females psychologically suffered more than males during the COVID-19 pandemic [10,11,12]; (2) young people were mentally more vulnerable than the elders [8,10,11]; (3) people with lower education levels [10,11] had greater depression and anxiety tendency; (4) health workers were more likely to have lower levels of sleep quality and higher levels of depression and anxiety than the others [8,14]; (5) people’s physical symptoms were negatively related to their psychological well-being [11,14]; (6) precautionary measures, such as wearing a mask, were associated with lower levels of anxiety and depression [11]; and (7) people who were unemployed tended to have lower levels of life satisfaction than the others [14].

### 1.2. Social Media and Mental Health

Social media became a major way to connect the Chinese general population, who stayed at home, to the outside world during the COVID-19 pandemic; people spent a considerable amount of time every day on social media, such as Wechat and Weibo. Studies have shown a substantial increase in the number of Weibo (Chinese Twitter) posts [9] and a higher level of social media exposure since the COVID-19 outbreak [7]. Higher levels of social media exposure were positively associated with the likelihood of anxiety and depression [7]. Remarkably, before the COVID-19 pandemic, studies showed mixed results on the relationship between social media use and mental health [16,17]. For example, Frison and colleagues [18] found the protective role of perceived social-media support in ameliorating the feeling of depression, while Park [19] demonstrated that greater depressive symptoms were associated with more social support on social media. A growing body of evidence suggested that the relationship between social media use and mental health was premised upon specific activities rather than the frequency of the activities [20,21,22,23]. It is valuable to understand the relationship between the general public’s specific social media activities and their mental health status during the COVID-19 pandemic.

### 1.3. Emotion-Regulation Strategies and Mental Health

Emotion-regulation strategies have a noteworthy effect on mental health [24,25]. Emotion-regulation refers to extrinsic and intrinsic processes that are responsible for monitoring, evaluating, and modifying emotional reactions [26]. Studies have shown that cognitive reappraisal and expressive suppression are the most common strategies among the numerous emotion-regulation strategies [27,28]. According to Ochsner and Gross [29], cognitive reappraisal may take two forms: (1) reinterpreting contextual aspects of stimuli and (2) distancing oneself from stimuli. In terms of the expressive suppression, one regulates emotional expression by controlling emotional behaviors [30].

Previous studies have shown a positive relationship between cognitive reappraisal and life satisfaction, and a negative relationship between expressive suppression and life satisfaction [31]. Moreover, cognitive reappraisal was negatively associated with anxiety and depression, while expressive suppression was positively related to anxiety and depression [32,33]. However, most of those studies focused on the direct relationships between emotion-regulation strategies and mental health, while none of them has considered the possible influences of emotion-regulation strategies on the relationships between general public’s diverse social media activities and mental health.

### 1.4. Theoretical Framework

To thoroughly explore the relationships between general public’s social media activities and mental health status, this study adopted Heaney and Israel’s [34] conceptual model of people’s social network/support and mental health status. The model illustrates that social networks and social support may have effects on an individual’s mental health in two possible ways (Figure 1). One (A) presents the direct effect of social networks and social support on mental health, and the other (B) presents a moderation effect of emotion-regulation strategies on the relationship between social networks/support and mental health.

### 1.5. Research Purpose

The purpose of this study is to investigate the framework shown in Figure 1, in order to achieve the following: (1) expand the understanding of relationships among social media activities, emotion-regulation strategies, and mental health, especially under the circumstance of COVID-19 pandemic; and (2) provide empirical evidence in assisting policy makers and mental-healthcare professionals to safeguard the psychological well-being of the general public, through social media, in the face of the worldwide pandemic. The specific research questions that guided this study are as follows:What are the relationships between the members of the general public’s varied social media activities and their mental health status (i.e., life satisfaction, sense of adequacy, depression, and anxiety) after the outbreak of COVID-19?What are the moderation effects, if any, of people’s emotion-regulation strategies on the relationships between their social media activities and mental health status?

## 2. Materials and Methods

### 2.1. Sample

Cross-sectional data were collected online through Wenjuanxing (a popular survey platform in China), from 19 to 21 February 2020, when the daily reported COVID-19 confirmed cases began to descend. Convenience and snowball sampling methods were used. The study was approved by the Academic Committee of East China Jiaotong University, to ensure the rights and welfare of the research participants were protected. Participation in this study was completely voluntary and anonymous. Overall, 4059 completed questionnaires were obtained, where 3159 (78%) of them were valid. Respondents aged ≥18 years were sampled from 32 provinces, municipalities, or special regions in China, with 51% females and 54% rural residents (vs. 25% from cities and 21% from towns). Most respondents (44%) had family incomes between 30,000 to 100,000 RMB (about 4392–14,640 USD, as of 10 September 2020), and 91% of them received or completed associate’s or bachelor’s levels of education.

### 2.2. Instrument

The survey questionnaire used in this study including items on participants’ demographic information, social media activities, emotional-regulation strategies, and psychological well-being (i.e., life satisfaction, sense of adequacy, depression, and anxiety measures). Self-designed, adapted, and existing questionnaire items were utilized. A pilot survey was carried out prior to our actual survey. A description of the measures is shown below. Appendix A
Table A1 provides detailed descriptive information of all the variables in this study.

#### 2.2.1. Life Satisfaction

Miao [35] designed the Multiple Happiness Questionnaire (MHQ) that measured nine aspects of people’s psychological well-being, with life satisfaction (including five items) being one of them. MHQ had been tested among university students and people aged ≥60 years old in China [35,36,37]. This study adopted the life satisfaction statements of the MHQ, but changed the original 7- to the current 5-point Likert scale. Factor analysis extracted 1 component and the reliability of this measurement was 0.91. The derived variable was created based on the mean of the items, with higher values indicating higher levels of life satisfaction.

#### 2.2.2. Sense of Adequacy, Depression, Anxiety

Based on Chan and Chan’s General Health Questionnaire (GHQ) (thirty questions) [38], Li and Boey [39] developed a GHQ version with twenty questions (GHQ-20) in three dimensions—sense of adequacy, depression, and anxiety. It had been tested among university students, prisoners, family caregivers, and the general population in China [39,40,41,42]. In our study, factor analysis on the three GHQ-20 dimensions extracted 1 component, respectively. The reliabilities were 0.77 (sense of adequacy), 0.65 (depression), and 0.81 (anxiety), respectively. The derived variables were created based on the sum of the items within each dimension.

#### 2.2.3. Social Media Activities

Self-designed items on social media activity variables were obtained based on the suggestions of experts in the related areas. Both COVID-19 related and the general social media activities were examined. The former included four aspects, such as COVID-19 information sharing, and the latter also contained four aspects, including social media dependence. Besides the dummy coded variables, the reliabilities were 0.84 (COVID-19 information sharing), 0.64 (COVID-19 online discussion), 0.72 (feeling toward COVID-19 information), and 0.76 (feeling toward social media interaction), respectively.

#### 2.2.4. Emotion-Regulation Strategies

Cognitive reappraisal and expressive suppression were measured as two emotion-regulation strategies in this study. Based on the two emotion-regulation questionnaires developed respectively by Liu [43] and Liu [44], as well as the study of Dixon-Gordon, Aldao, and Reyes [45], cognitive reappraisal contained ten statements and expressive suppression contained five statements. Factor analysis on the two strategies extracted 1 component, respectively. The reliability for cognitive reappraisal was 0.94 and for expressive suppression was 0.88. The derived variables were created based on the mean of the items within each dimension.

### 2.3. Data Analysis

Through using SPSS 26, both descriptive and inferential analyses were adopted in this research. The descriptive analyses examined the general public’s mental health status, as well as the relationships among social media activities, emotion-regulation strategies, and psychological well-being without controlling for all the other effects. To further our understanding, multiple linear regressions were applied to examine the relationships between social media activities and people’s mental health along with the consideration of the moderation effects of emotion-regulation strategies.

## 3. Results

### 3.1. Descriptive Statistics and Correlations

At the time of our survey, over half of the participants reported that they used the Internet more than six hours per day, and over half of them reported that they spent at least 30% of their time online, looking at news and information about the COVID-19 pandemic. This suggested that online surfing, as well as concerns of the pandemic, became a significant part of the general population’s everyday life.

Our results indicated 2.7 percent of the respondents showed a low life-satisfaction level (2 SD below the mean); 5.1 percent showed a low sense of adequacy level (2 SD below the mean); 6.8 percent presented a high depression level (2 SD above the mean); and 7.4 percent presented a high anxiety level (2 SD above the mean). Thus, relatively fewer people felt unsatisfied about their life, slightly more felt inadequate, even more had a high depression tendency, while anxiety seemed to be the most prevalent disorder. Comparing with Gao et al.’s [7] research that analyzed the data collected at the early stage (31 January to 2 February) of the COVID-19, the prevalence of depression and anxiety in our study was largely reduced (Gao et al.’s research: depression, 48.3 percent; anxiety, 22.6 percent), though, admittedly, different standards were applied to identify depression and anxiety.

Table 1 shows that life satisfaction and sense of adequacy are moderately and positively related (r = 0.53); similarly, depression and anxiety are moderately and positively related (r = 0.45). The relationships between positive (satisfaction and sense of adequacy) and negative mental health indicators (depression and anxiety) are all negative and in the magnitudes between weak to moderate, with r ranging from −0.25 to −0.37.

All social media activity variables presented positive and weak-to-moderate relationships with positive psychological indicators, with r ranging from 0.12 to 0.36. To put it differently, without considering any other effects, higher levels of our measured social media activities tended to be associated with higher levels of people’s psychological well-being. Meanwhile, excepting the negative and weak associations of feeling toward COVID-19 information (r = −0.17) and feeling toward social media interaction (r = −0.16) with depression, the magnitudes of the relationships between social media activities and negative psychological indicators were close to none (r ranging from 0.002 to −0.09).

In terms of the social media activity dummy variables (Table 2), without holding constant the other factors, comparing with people who did not feel attached to social media (social media belongingness), people who felt attached presented no significant difference in their life satisfaction (mean gap: 0.02), a lower level of sense of adequacy (mean gap: −4.09), a higher level of depression (mean gap: 0.23), and also a higher level of anxiety (mean gap: 0.43). Comparing with people who did not addicted to social media (social media dependence), people who felt addicted had lower levels of life satisfaction (mean gap: −0.17) and sense of adequacy (mean gap: −0.97), but higher levels of depression (mean gap: 0.38) and anxiety (mean gap: 0.42). Moreover, comparing with people who did not tend to stop what they regarded as inappropriate comments/opinions, people who tended to stop had higher levels of life satisfaction (mean gap: 0.30) and sense of adequacy (mean gap: 0.60), but lower levels of depression (mean gap: −0.09), though no disparity was found in anxiety (mean gap: −0.06).

Cognitive reappraisal was positively and weakly to moderately associated with life satisfaction (r = 0.31) and sense of adequacy (r = 0.29). In contrast, it was negatively and weakly associated with depression (r = −0.23), while it showed no relationship with anxiety (r = −0.09). In the meantime, expressive suppression was positively but weakly associated with life satisfaction (r = 0.11), while it basically presented no relationship with sense of adequacy (r = 0.04). With considering the magnitude of the relationships, expressive suppression strategy presented no relationships with depression (r = −0.05) and anxiety (r = 0.02). In general, cognitive reappraisal, rather than expressive suppression, had stronger associations with the general public’s mental health.

### 3.2. Relationships between Social Media Activities and Mental Health Status

Social Media Activities and Life Satisfaction (Table 3): When holding constant the other variables, COVID-19 online discussion and social media judgement had significant negative relationships with life satisfaction (β = −0.295, *p* < 0.01; β = −0.395, *p* < 0.01, respectively). A one-unit increase in COVID-19 online discussion was associated with a 0.295 points decrease in life satisfaction. Moreover, people who tended to judge other people’s behaviors on social media generally had lower levels of life satisfaction than those who did not judge. In comparison, higher levels of positive COVID-19 information sharing and positive feelings toward COVID-19 information were significantly, though marginally, related to people’s higher life-satisfaction levels (β = 0.189, *p* = 0.05; β = −0.395, *p* = 0.05, respectively). All the other social media activity variables presented no significant relationships with life satisfaction. Those results indicated that, during the COVID-19 pandemic, people who were more likely to share positive information about COVID-19 held more positive feelings toward COVID-19 information, discussed less about COVID-19 with others (e.g., the current COVID-19 situation), and judged less when using the social media tended to have higher levels of life satisfaction. The model, in total, explained 30.5% of the variance in life satisfaction.

Social Media Activities and Sense of Adequacy (Table 3): After controlling for all the other variables, COVID-19 information sharing had a significantly positive relationship with people’s sense of adequacy (β = 0.930, *p* < 0.01). A one-unit increase in COVID-19 information sharing was associated with a 0.930 units increase in sense of adequacy. In contrast, social media dependence exhibited a marginal significant negative relationship with sense of adequacy (β = −1.090, *p* = 0.01). That is, people who had a dependence on the internet generally had a lower level of sense of adequacy than those who did not have such a dependence. The other social media activities showed no significant relationships with sense of adequacy. The results indicated that, during the COVID-19 pandemic, people who shared more positive COVID-19 information and were less dependent on the internet tended to have higher levels of sense of adequacy. This model explained 27.8% of the total variance in sense of adequacy.

Social Media Activities and Depression (Table 3): When holding constant the other variables, social media dependence and social media self-expression had significantly positive relationships with depression (β = 0.776, *p* < 0.01; β = 0.340, *p* < 0.05, respectively). More specifically, people who had a dependence on internet generally had a higher level of depression than those who did not have such dependence. A one-unit increase in social media self-expression was associated a 0.340 units increase in depression. In comparison, COVID-19 information sharing and feeling toward social media interaction had significantly negative relationships with depression (β = −0.340, *p* < 0.05; β = −0.833, *p* < 0.001, respectively). A one-unit increase in COVID-19 information sharing and feeling toward social media interaction was associated with a 0.340 and 0.833 units decrease in depression, respectively. The rest of the social-media-activity variables showed no significant relationships with depression. In general, during the COVID-19 pandemic, people who shared more positive COVID-19 information held more positive feelings toward social media interaction, did not have a high dependence on the internet, and were less likely to express themselves through the social media tended to have lower depression levels. The model, in total, explained 15.3% of the variance in depression.

Social Media Activities and Anxiety (Table 3): With everything else being equal, COVID-19 online discussion had a significant positive relationship with anxiety (β = 0.804, *p* < 0.001). A one-unit increase in COVID-19 online discussion was associated with a 0.804 units increase in anxiety. Meanwhile, COVID-19 information sharing, feeling toward COVID-19 information, and feeling toward social media interaction had significant negative relationships with anxiety (β = −0.454, *p* < 0.05; β = −0.365, *p* < 0.05; β = −0.630, *p* < 0.01, respectively). A one-unit increase in COVID-19 information sharing, feeling toward COVID-19 information, and feeling toward social media interaction was associated with a 0.454, 0.365, and 0.630 units decrease in anxiety, respectively. The other social media activities presented no significant relationships with anxiety. In general, during the COVID-19 pandemic, people who discussed less about the COVID-19, shared more positive information, and held more positive feelings toward the COVID-19 information and social media interaction, tended to have lower levels of anxiety. The model, in total, explained 8.9% of the variance in anxiety, which was the lowest among the four mental health indicators (i.e., life satisfaction, sense of adequacy, depression, and anxiety) we examined.

### 3.3. Relationships between Emotion-Regulation Strategies and Mental Health Status

After controlling for all the other variables, cognitive reappraisal had significant positive relationships with life satisfaction (β = 0.254, *p* < 0.05) and sense of adequacy (β = 1.396, *p* < 0.001), but significant negative relationships with depression (β = −1.753, *p* < 0.001) and anxiety (β = −0.671, *p* < 0.01) (Table 3). That is, higher levels of cognitive reappraisal were related to higher levels of people’s life satisfaction and sense of adequacy. Meanwhile, expressive suppression was negatively related to people’s sense of adequacy (β = −1.225, *p* < 0.001), but positive related to depression (β = 1.028, *p* < 0.001). Interestingly, expressive suppression was not associated with life satisfaction (β = −0.133, *p* > 0.05) and anxiety (β = 0.335, *p* > 0.05). Thus, higher levels of expressive suppression were related to people’s lower levels of sense of adequacy and higher levels of depression.

### 3.4. Interactive Associations between Emotion-Regulation Strategies and Social Media Activities

As shown in Table 4, with everything else being equal, among people with higher levels of cognitive reappraisal, the negative relationship between social media judgement and life satisfaction was reduced and even close to none (β = 0.113, *p* < 0.05). The positive relationship between COVID-19 information sharing and sense of adequacy was somewhat decreased in people with higher cognitive appraisal levels and even close to none (β = −0.295, *p* < 0.01). The negative associations of feeling toward COVID-19 information (β = 0.135, *p* < 0.05) and feeling toward social media interaction (β = 0.328, *p* < 0.001) with depression were reduced and even became positive when people holding a higher cognitive reappraisal score; meanwhile, the positive associations of social media dependence (β = −0.225, *p* < 0.05) and social media self-expression (β = −0.133, *p* < 0.05) with depression became weaker and even reached none. Last, the positive associations of COVID-19 online discussion (β = −0.193, *p* < 0.05) and social media belongingness (β = −0.277, *p* < 0.05) with anxiety were diminished and even close to none for people with higher cognitive appraisal levels; in comparison, the negative relationship between feeling toward social media interaction and anxiety was decreased (β = 0.165, *p* = 0.05).

On the other hand, this study did not find any significant interactions between social media activities and expressive suppression on life satisfaction and sense of adequacy. Interaction results indicated among people with higher levels of expressive suppression, the positive relationship between COVID-19 online discussion and depression was reduced and even turned into a negative relationship (β = −0.134, *p* < 0.05), while the negative relationship between feeling toward COVID-19 information and depression became stronger (β = −0.096, *p* = 0.05). Finally, the positive association of social media belongingness with anxiety became stronger among people with higher expressive-suppression levels (β = 0.250, *p* < 0.05).

Figure 2 presents selected significant interactions. The relationships between social media activities and mental health indicators in people with lower levels of cognitive reappraisal were in the same directions as those relationships without considering the moderators; meanwhile, in people with higher levels of cognitive reappraisal, such relationships tended to be much weaker or none. Generally speaking, embracing higher levels of cognitive reappraisal could ameliorate the adverse relationships between certain social media activities and mental health indicators. For example, among people in the high-cognitive-reappraisal group, social media judgment (Figure 2a) and COVID-19 online discussion (Figure 2b) basically exhibited no relationships with life satisfaction and anxiety. In comparison, among people in the low-cognitive-reappraisal group, those who tended to be judgmental on social media presented lower life-satisfaction levels than those who did not; moreover, people who discussed more about COVID-19 on social media presented higher anxiety levels than those who did not. Remarkably, for people in the low-cognitive-reappraisal group, sharing more positive information of COVID-19 on social media might increase their sense of adequacy (Figure 2c), and having more positive feelings toward social media interactions with other people might lower their level of depression (Figure 2d).

## 4. Discussion

### 4.1. Social Media Activities and General Population’s Mental Health after the COVID-19 Outbreak

A key merit for using social media is that it can provide valuable peer, social, and emotional supports for the general public [46,47,48], especially under the strict “staying at home” policy in China. However, the findings of this study revealed that engaging in social media communication was not necessarily associated with better mental health status for the general population, which was inconsistent with Fergie and colleague’s [49] suggestion. Our results showed that the way of using social media mattered. For example, people who were more likely to share positive information of COVID-19 on social media tended to have higher levels of life satisfaction and sense of adequacy, as well as lower levels of depression and anxiety; meanwhile, people who were inclined to be involved in discussions about the COVID-19 on social media tended to have higher anxiety and lower life satisfaction levels. The results added empirical evidence to the explanations of the mixed results in the relationships between social media activities and people’s mental health in prior studies [20]. That is, different ways of using social media can yield varied impacts on people’s psychological well-being. Meanwhile, people’s attitudes toward the communication of COVID-19 information seemed to play a significant role under the current circumstances. The possible explanation was that, during the pandemic, people may bear fear and other negative emotions, which may contaminate the other ones through certain social media activities [9,10].

Our results also indicated that people who had a higher tendency of expressing themselves and were addicted to social media were more likely to have higher levels of depression; meanwhile, people who held positive feelings toward social media interactions tended to have lower levels of depression and anxiety. In addition, people who tended to judge others on social media were more likely to have lower life-satisfaction levels. Those results on social media activities were consistent with previous studies on social media dependence [50], self-expression [51], and supports from social media [48]. Our findings echoed Heaney and Israel’s [34] framework of social networks and social support. That is, the general public’s social media activities could have direct effects on their mental health status. On the other hand, our study found some unique properties, in which people’s attitudes and feelings toward the social media activities did matter under the circumstances of COVID-19 pandemic.

### 4.2. Emotional-Regulation Strategies and Their Moderation Effects

The relationships between cognitive reappraisal/expressive suppression and people’s mental health were consistent with Hu, Zhang, and Wang’s [52] meta-analysis, in which they found cognitive reappraisal positively associated with life satisfaction, and negatively associated with anxiety and depression; meanwhile, expressive suppression was positively related to negative mental health indicators and negatively related to positive mental health indicators. Furthermore, the moderation effects found in this study revealed that people with lower levels of cognitive reappraisal were more vulnerable in the relationships between social media activities and mental health. In contrast, possessing higher levels of cognitive reappraisal may benefit the general public by reducing the adverse influence of social media activities on their mental health. The findings provided empirical evidence on the importance of cognitive reappraisal on people’s mental health status when dealing with social media activities, especially when the general public was under the stress resulted from the COVID-19 pandemic. Under such circumstances, the emotion-regulation strategies may matter more than the actual social media activities.

## 5. Conclusions

The findings of this study provided insights on general public’s social media activities and their mental health during the COVID-19 pandemic. It expanded the theory of how social support/network can associate with human being’s mental health and how it can interact with emotion-regulation strategies. The findings suggest that researchers who study the relationships between people’s social media use and mental health status might need to consider their emotional-regulation strategies as an important influential factor. Furthermore, the findings suggest governments, policy makers, practitioners, and related others should pay more attention to the general public’s social media literacy, for example, how to distinguish rumors and reliable information, and share the information wisely. Moreover, the media were encouraged to share more suggestions of positive mental health practices during the pandemic, to help the public overcome any negative feelings caused by the pandemic.

Future studies are suggested to (1) examine the relationships among social media use, emotion-regulation strategies, and mental health status, in broader contexts, for example, in countries with different systems; (2) use longitudinal data to investigate the relationships to reach more reliable results and reveal any shift of the relationships along the time; and (3) use various sources of data, other than self-reported measures, to examine the validity of the relationships found in this study.

## 6. Limitations

This study is subject to several limitations. First, people’s mental health and social media activities were measured by self-report survey, which may not fully reflect the true status. Second, the cross-sectional data used in this study limited the estimation of the effects of COVID-19 on people’s psychological well-being and its relation to social media practices. Third, this study was carried out in mainland China, with its own context; thus, the findings might not be generalized to other social and cultural contexts.

## Figures and Tables

**Figure 1 ijerph-17-08931-f001:**
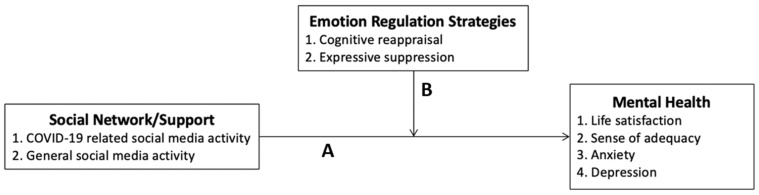
The conceptual framework of relationships among people’s social media activities, emotion-regulation strategies, and mental health status.

**Figure 2 ijerph-17-08931-f002:**
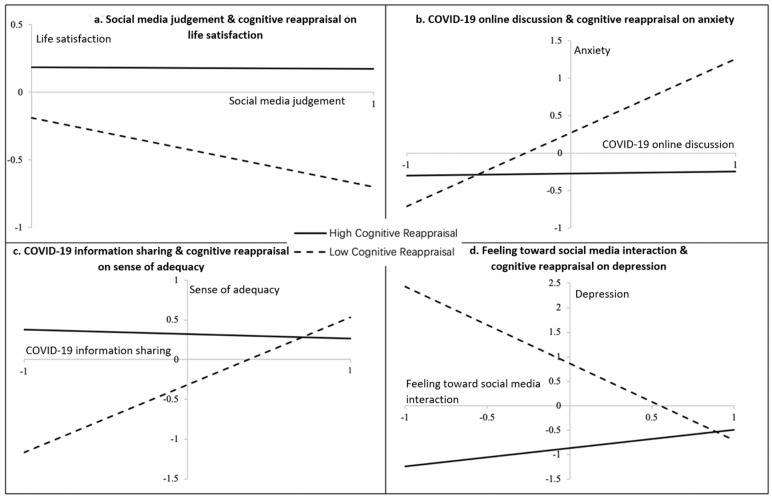
Selected interactions between social media activities and cognitive reappraisal on mental health. Notes: (1) Graphs were plotted by using standardized coefficients. For dummy variables, the two groups were divided by yes (1) or no (0). The high-cognitive-reappraisal regulation group was set at one SD above the mean, while the low group was set at one SD below the mean. (2) Full interaction graphs are available upon request.

**Table 1 ijerph-17-08931-t001:** Correlations between psychological well-being and social media variables.

		Psychological Well-Being				Social Media Variables			
		1	2	3	4	5	6	7	8
**Psychological well-being**									
Life satisfaction	1								
Sense of adequacy	2	0.530 ***							
Depression	3	−0.326 ***	−0.369 ***						
Anxiety	4	−0.250 ***	−0.284 ***	0.448 ***					
**Social media variables**									
COVID-19 information sharing	5	0.280 ***	0.247 ***	−0.077 ***	−0.015				
COVID-19 online discussion	6	0.246 ***	0.181 ***	−0.078 ***	0.011	0.598 ***			
Feeling toward COVID-19 information	7	0.338 ***	0.269 ***	−0.166 ***	−0.093 ***	0.347 ***	0.354 ***		
Feeling toward COVID-19 interaction	8	0.362 ***	0.289 ***	−0.159 ***	−0.054 **	0.499 ***	0.611 ***	0.430 ***	
Social media self-expression	9	0.170 ***	0.119 ***	−0.031	0.002	0.476 ***	0.422 ***	0.180 ***	0.413 ***

Notes: * *p* < 0.05, ** *p* < 0.01, and *** *p* < 0.001. A correlation table with all variables included is available upon request.

**Table 2 ijerph-17-08931-t002:** Mean comparison of mental health indicators on social media activities.

		Life Satisfaction	Sense of Adequacy	Depression	Anxiety
Social media belongingness	No	3.343	6.103	0.665	0.634
	Yes	3.358	5.694	0.897	1.064
	Yes–No	0.015	−0.409 ***	0.232 ***	0.430 ***
Social media dependence	No	3.390	6.207	0.647	0.668
	Yes	3.214	5.242	1.026	1.091
	Yes–No	−0.176 ***	−0.965 ***	0.379 ***	0.423 ***
Social media judgment	No	3.207	5.694	0.781	0.799
	Yes	3.509	6.294	0.690	0.738
	Yes–No	0.302 ***	0.600 ***	−0.091 *	−0.061

Notes: * *p*< 0.05, ** *p* < 0.01, and *** *p* < 0.001.

**Table 3 ijerph-17-08931-t003:** Regression results on life satisfaction, sense of adequacy, depression, and anxiety.

	Life Satisfaction		Sense of Adequacy		Depression		Anxiety	
	β (*se*)		β (*se*)		β (*se*)		β (*se*)	
Social media variables								
COVID-19 information sharing	0.189 ~	(0.097)	0.930 **	(0.315)	−0.340 *	(0.160)	−0.454 *	(0.201)
COVID-19 online discussion	−0.295 **	(0.112)	−0.509	(0.365)	0.298	(0.185)	0.804 ***	(0.233)
Feeling toward COVID-19 information	0.164 ~	(0.086)	0.217	(0.280)	-0.196	(0.142)	−0.365 *	(0.178)
Feeling toward social media interaction	0.186	(0.113)	0.092	(0.368)	−0.883 ***	(0.187)	−0.630 **	(0.235)
Social media belongingness	−0.007	(0.166)	−0.024	(0.539)	0.305	(0.273)	0.420	(0.344)
Social media dependence	−0.031	(0.174)	−1.090 ~	(0.567)	0.776 **	(0.288)	0.121	(0.362)
Social media self-expression	0.016	(0.092)	−0.174	(0.298)	0.340 *	(0.151)	0.149	(0.190)
Social media judgment	−0.395 **	(0.149)	−0.246	(0.485)	0.388	(0.246)	0.382	(0.309)
Emotion-Regulation Strategies								
Cognitive reappraisal	0.254 *	(0.117)	1.396 ***	(0.381)	−1.753 ***	(0.193)	−0.671 **	(0.243)
Expressive suppression	−0.133	(0.112)	−1.225 ***	(0.363)	1.028 ***	(0.184)	0.335	(0.231)
Adjusted R^2^	30.5%		27.8%		15.3%		8.9%	

Notes: ~ *p* = 0.05, * *p* < 0.05, ** *p* < 0.01, and *** *p* < 0.001. Results of the background and other control variables (e.g., gender, age, and education level) were not reported in this table. Full regression table is available upon request.

**Table 4 ijerph-17-08931-t004:** Interaction results on life satisfaction, sense of adequacy, depression, and anxiety.

	Life Satisfaction		Sense of Adequacy		Depression		Anxiety	
	β (*se*)		β (*se*)		β (*se*)		β (*se*)	
**Social media variables x Cognitive reappraisal**								
COVID-19 information sharing	−0.029	(0.034)	−0.295 **	(0.112)	0.083	(0.057)	0.128	(0.071)
COVID-19 online discussion	0.074	(0.041)	0.170	(0.132)	0.042	(0.067)	−0.193 *	(0.084)
Feeling toward COVID-19 information	−0.043	(0.034)	−0.058	(0.110)	0.135 *	(0.056)	0.106	(0.070)
Feeling toward social media interaction	−0.028	(0.042)	−0.082	(0.137)	0.328 ***	(0.069)	0.165~	(0.087)
Social media belongingness	0.043	(0.062)	0.114	(0.201)	−0.184	(0.102)	−0.277 *	(0.128)
Social media dependence	−0.067	(0.067)	−0.108	(0.216)	−0.225 *	(0.110)	0.217	(0.138)
Social media self-expression	0.011	(0.034)	0.069	(0.111)	−0.133 *	(0.056)	−0.054	(0.071)
Social media judgment	0.113 *	(0.053)	0.057	(0.173)	−0.112	(0.088)	−0.125	(0.110)
**Social media variables x Expressive suppression**								
COVID-19 information sharing	−0.014	(0.030)	0.062	(0.097)	0.027	(0.049)	0.010	(0.062)
COVID-19 online discussion	0.007	(0.036)	−0.065	(0.116)	−0.134 *	(0.059)	−0.035	(0.074)
Feeling toward COVID-19 information	0.017	(0.031)	0.041	(0.100)	−0.096 ~	(0.051)	−0.026	(0.064)
Feeling toward COVID-19 interaction	0.022	(0.039)	0.201	(0.126)	−0.104	(0.064)	−0.010	(0.080)
Social media belongingness	−0.053	(0.057)	−0.240	(0.185)	0.148	(0.094)	0.250 *	(0.118)
Social media dependence	0.050	(0.063)	0.297	(0.205)	0.054	(0.104)	−0.187	(0.131)
Social media self-expression	−0.011	(0.031)	0.005	(0.101)	0.042	(0.051)	0.022	(0.064)
Social media judgment	0.022	(0.048)	−0.007	(0.155)	0.026	(0.078)	0.016	(0.099)

Notes: (1) ~ *p* = 0.05, * *p* < 0.05, ** *p* < 0.01, and *** *p* < 0.001. (2) Cognitive reappraisal and expressive suppression were the two components of the emotion-regulation strategies being examined in this study. (3) Interaction results based on regression analyses in Table 3.

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
