# Peer review of "Social Media Activities, Emotion Regulation Strategies, and Their Interactions on People’s Mental Health in COVID-19 Pandemic"

_ijerph, 2020, doi:10.3390/ijerph17238931_

Round 1

Reviewer 1 Report

It is an interesting article with strong theoretical framework. It is timely study for COVID-19 pandemic situation.

Analysis and result did not have any flaw.

Discussion is also acceptably written.

Therefore it should be published quickly.

Author Response

Responses: We appreciate the efforts of reviewer #1 and very thanks for the positive comments.

Revisions: Not applicable.

Reviewer 2 Report

This manuscript addresses an important and timely topic of the relationship between mental health and social media under the ongoing pandemic.

The Introduction paragraph can be improved by adding the article's specific thesis (for example, by including a short rephrasing of what is mentioned in Section 1.5).

The survey period of 3 days in late February seems remarkably short and it is questionable how much this very brief snapshot can tell us about the long-term nature of mental health and its relation to social media practices. In addition, anonymous (or even non-anonymized) self-reporting regarding both mental health and social media use is always questionable in terms of its accuracy.

One key weakness with the current study is that we are only given a snapshot of respondents' mental health during the pandemic, but we do not know what the respondents' mental health state was prior to the pandemic. Without longitudinal data on individuals it is hard to estimate the actual effects of a) the pandemic and b) the use of social media during the pandemic.

One open question in this study is what is particular about the Covid-19 pandemic? More precisely, the study analyzed the practice of information sharing related to the pandemic. However, would there be similar results for other kinds of disasters/issues, such as earthquakes, floods, political instability, economic crises, etc.? Is there any previous literature that the authors could draw from regarding social media use and disasters?

The suggestion in the conclusion that "governments, policy makers, practitioners, and related others should pay more attention to help the general public establish positive attitudes and emotion regulation strategies through, for example, social media interactions, with the aim of boosting people’s psychological well-being" is very vague and sounds somewhat problematic. What exactly do they recommend these agents to do? It sounds like a recommendation for large-scale social-media based social engineering. While I support the idea of improving psychological well-being, top-down interventions aimed at manipulating citizens' psychology could have worrisome implications for large-scale psychological control. In the context of mainland China this kind of expectation of state-driven emotional control might be accepted or even expected, but in other societies the focus would be rather on education of citizens to better equip them with social media literacy and positive mental health practices.

Also, the conclusion is rather short. The authors could expand this by discussing broader implications of their research and future studies which should be conducted to further validate and deepen their findings.

Author Response

  1. The Introduction paragraph can be improved by adding the article's specific thesis (for example, by including a short rephrasing of what is mentioned in Section 1.5).

    Responses: We added the specific thesis of this study at the end of the first paragraph in Introduction.

    Revisions: Please refer to line 63-65 on page 2.
  2. The survey period of 3 days in late February seems remarkably short and it is questionable how much this very brief snapshot can tell us about the long-term nature of mental health and its relation to social media practices. In addition, anonymous (or even non-anonymized) self-reporting regarding both mental health and social media use is always questionable in terms of its accuracy.

    Responses: We acknowledged the weakness of our cross-sectional data and self-report survey. We added this information in a new paragraph titled Limitation.

    Revisions: Please refer to the Limitation on page 11-12.
  3. One key weakness with the current study is that we are only given a snapshot of respondents' mental health during the pandemic, but we do not know what the respondents' mental health state was prior to the pandemic. Without longitudinal data on individuals it is hard to estimate the actual effects of a) the pandemic and b) the use of social media during the pandemic.

    Responses: We acknowledged the weakness of using cross-sectional data in this study. However, we had clues from summary of the mixed results of Chinese people’s mental health status before and after the pandemic. Most studies found worse mental health status after the pandemic (line 71-78).

    Revisions: Please refer to the Limitation on page 11-12.
  4. One open question in this study is what is particular about the Covid-19 pandemic? More precisely, the study analyzed the practice of information sharing related to the pandemic. However, would there be similar results for other kinds of disasters/issues, such as earthquakes, floods, political instability, economic crises, etc.? Is there any previous literature that the authors could draw from regarding social media use and disasters?

    Responses: We added some information of the characteristics of the COVID-19 pandemic in China. It was the first time in which Chinese government took such massive lockdown policy. Also, we have reviewed some studies of social media use and crisis, e.g., one similar result was that the frequency of social media use increased during the crisis. However, these studies had little relationship with the current study, especially in the context of China. We referred to one study of media, mental health, and H1N1 pandemic, which might relate closely to our research.

    Revisions: Please refer to line 38-39 on page 1, and line 59-61 on page 2.
  5. The suggestion in the conclusion that "governments, policy makers, practitioners, and related others should pay more attention to help the general public establish positive attitudes and emotion regulation strategies through, for example, social media interactions, with the aim of boosting people’s psychological well-being" is very vague and sounds somewhat problematic. What exactly do they recommend these agents to do? It sounds like a recommendation for large-scale social-media based social engineering. While I support the idea of improving psychological well-being, top-down interventions aimed at manipulating citizens' psychology could have worrisome implications for large-scale psychological control. In the context of mainland China this kind of expectation of state-driven emotional control might be accepted or even expected, but in other societies the focus would be rather on education of citizens to better equip them with social media literacy and positive mental health practices.

    Responses: We agreed with the reviewer, and rewrote the suggestion to make it clear and practical. Also, we acknowledged the limitation of social cultural context of China in our study.

    Revisions: Please refer to the changes in Conclusion on page 11 and Limitation on page 11-12.
  6. Also, the conclusion is rather short. The authors could expand this by discussing broader implications of their research and future studies which should be conducted to further validate and deepen their findings.

    Responses: We added more implications of our study and suggestions to future research in Conclusion.

    Revisions: Please refer to the changes in Conclusion on page 11.

Reviewer 3 Report

The work is frankly well thought out and provides new relevant information with clear implications and applications for the society that is living the COVID-19 outbreak. 

I found its rationale or background very precise, as well as the formulation of objectives and hypotheses planning the contrast of Heaney and Israel's model and its inclusion of figure 1.

There are only a few minor issues that need to be clarified: 

1. Line 29. The summary lacks a final sentence that points out the main contribution or conclusion of the work. 
2. Lines 145-171. In the description of the internal consistencies of the assessment instruments, intervals of consistencies are included. It is not appropriate, if the measure has three dimensions and there are three consistency values, they should all be included specifically and pointing out their properties. It is true that all this information is in the appendix, but in that case, the appendix could be included as a fifth table in the work itself, not as appendix. 
3. Lines 195-196. Using the cut-off point of more than 2 DT is arbitrary, it is true that when making comparisons the authors indicate it as limitation, so I think it could be considered right. 

4. Lines 293-296 and 320-322. Tables 3 and 4 have a problem. Social Media Variables and Emotion Regulations Strategies are included in both, but a heading equivalent to "Social Media Variables" on the top is not used in column 1 for the regulation strategies (which would be before cognitive reappraisal and expressive suppression). In addition, from Seek Psychological Assistance to Health workers (vs no role) variable would be another group of variables which, despite being included, are not discussed at any time, are not in the objectives of the study, etc. Consequently, they are taken into account to calculate r2, but are not commented on. This should be eliminated or on the contrary included in the work with all the consequences that this would have. 

5. Finally, the interactive associations are very interesting and suggestive, but perhaps they deserve a little more explanation in the section on data analysis, since they are less well known than the regressions. 

6. A last comment is only a suggestion or recommendation to keep in mind. Concerning the mentioned model in Figure 1, when reading the model in Figure 1 the first thought coming to mind is that the authors will propose a model of moderation through Process, however regressions and interactions are provided. The authors have thought to add to the work the data of this model of moderation, although it is understood that it would be too long, they could contribute it for the four variables that are presented in figure 2, including emotional regulation as moderator, mental health (satisfaction, anxiety, depression and adequacy) as criterion variable and the use of social networks as predictor variable. A more rounded work would remain.

Anyway, congratulations for such a good job!

Author Response

1. Line 29. The summary lacks a final sentence that points out the main contribution or conclusion of the work.

Responses: We added a sentence to point out the contribution.

Revisions: Please refer to line 30-32 in Abstract.

2. Lines 145-171. In the description of the internal consistencies of the assessment instruments, intervals of consistencies are included. It is not appropriate, if the measure has three dimensions and there are three consistency values, they should all be included specifically and pointing out their properties. It is true that all this information is in the appendix, but in that case, the appendix could be included as a fifth table in the work itself, not as appendix.

Responses: We revised this by providing individual Cronbach’s alpha values.  

Revisions: Please refer to the line 185-186 and line 192-194 on page 4.

3. Lines 195-196. Using the cut-off point of more than 2 DT is arbitrary, it is true that when making comparisons the authors indicate it as limitation, so I think it could be considered right.

Responses: Thanks for pointing it out.

Revisions: Not applicable.

4. Lines 293-296 and 320-322. Tables 3 and 4 have a problem. Social Media Variables and Emotion Regulations Strategies are included in both, but a heading equivalent to "Social Media Variables" on the top is not used in column 1 for the regulation strategies (which would be before cognitive reappraisal and expressive suppression). In addition, from Seek Psychological Assistance to Health workers (vs no role) variable would be another group of variables which, despite being included, are not discussed at any time, are not in the objectives of the study, etc. Consequently, they are taken into account to calculate r2, but are not commented on. This should be eliminated or on the contrary included in the work with all the consequences that this would have.

Responses: We 1) added “Emotion Regulation Strategies”in Table 3; 2) removed the variables that were not the major independent variables from table 3 ; and 3) added a note to Table 4.

Revisions: Please refer to the Table 3 on page 8 and Table 4 on page 9.

5. Finally, the interactive associations are very interesting and suggestive, but perhaps they deserve a little more explanation in the section on data analysis, since they are less well known than the regressions. 

Responses: We added more explanation to clarify the findings of interaction.

Revisions: Please refer to line 342-375 on page 8-9.

6. A last comment is only a suggestion or recommendation to keep in mind. Concerning the mentioned model in Figure 1, when reading the model in Figure 1 the first thought coming to mind is that the authors will propose a model of moderation through Process, however regressions and interactions are provided. The authors have thought to add to the work the data of this model of moderation, although it is understood that it would be too long, they could contribute it for the four variables that are presented in figure 2, including emotional regulation as moderator, mental health (satisfaction, anxiety, depression and adequacy) as criterion variable and the use of social networks as predictor variable. A more rounded work would remain.

Responses: We modified Figure 1 and its description in order to reduce the possible misunderstanding.

Revisions: Please refer to Figure 1 and its description (line 127-128, 135) on page 3.

Round 2

Reviewer 2 Report

The authors have done well in revising the paper, and their revisions have answered my concerns. While some of the study limitations cannot be addressed because of the structural limitations of the original research, the authors have done a good job in highlighting what the study can offer and they have made the limitations clear. After some language editing, I believe that the paper is ready to be published.

One minor note: In Section 2.2 (Instruments), they should have the paragraph headings for each variable (e.g., Life Satisfaction) on a separate line as a sub-heading to make it clear. Or, use colons after the variables (e.g., Line 153 could read, "Life Satisfaction: Miao..."

Author Response

Thanks again for the thoughtful comments. We made changes according to the comments and marked them in red. We also made the changes of the 1st round review back into black. So only the changes in the 2nd round review were shown.

1. One minor note: In Section 2.2 (Instruments), they should have the paragraph headings for each variable (e.g., Life Satisfaction) on a separate line as a sub-heading to make it clear. Or, use colons after the variables (e.g., Line 153 could read, "Life Satisfaction: Miao..."

Responses: We agreed with the reviewer, and put the variable names as sub-headings on a separate line.

Revisions: Please refer to the changes on page 4 and 5.